# The Diabetic Cardiorenal Nexus

**DOI:** 10.3390/ijms23137351

**Published:** 2022-07-01

**Authors:** John A. D’Elia, George P. Bayliss, Larry A. Weinrauch

**Affiliations:** 1Kidney and Hypertension Section, E P Joslin Research Laboratory, Joslin Diabetes Center, Boston, MA 02215, USA; 2Division of Organ Transplantation, Rhode Island Hospital, Providence, RI 02903, USA; gbayliss@lifespan.org

**Keywords:** heart failure, kidney disease, cardiorenal syndrome

## Abstract

The end-stage of the clinical combination of heart failure and kidney disease has become known as cardiorenal syndrome. Adverse consequences related to diabetes, hyperlipidemia, obesity, hypertension and renal impairment on cardiovascular function, morbidity and mortality are well known. Guidelines for the treatment of these risk factors have led to the improved prognosis of patients with coronary artery disease and reduced ejection fraction. Heart failure hospital admissions and readmission often occur, however, in the presence of metabolic, renal dysfunction and relatively preserved systolic function. In this domain, few advances have been described. Diabetes, kidney and cardiac dysfunction act synergistically to magnify healthcare costs. Current therapy relies on improving hemodynamic factors destructive to both the heart and kidney. We consider that additional hemodynamic solutions may be limited without the use of animal models focusing on the cardiomyocyte, nephron and extracellular matrices. We review herein potential common pathophysiologic targets for treatment to prevent and ameliorate this syndrome.

## 1. Introduction

In the latter stages of heart failure or kidney disease, the interdependence of these two organ systems has become known as cardiorenal syndrome. Although some mechanisms of the kidney (purely nephritic, obstruction or genetic) and heart muscle (myocarditis, valvular or genetic causes) dysfunction may be independent of each other, others are dependent. Understanding mechanisms common to injuries of both organ systems has led to critical advances. A review of commonalities, linking renal and myocardial cellular and organ dysfunction will be helpful. We examine mechanisms that contribute to the pathogenesis of myocardial dysfunction with pathophysiologic contributions of obesity, diabetes and kidney disease. Mechanisms of hemodynamic overload, ischemia-related dysfunction, ventricular remodeling, excessive neuro-humoral stimulation, abnormal myocyte calcium cycling, cytokine-induced proliferation of extracellular matrix and accelerated apoptosis are within the scope of this review.

Heart failure in patients with diabetic kidney disease is associated with interstitial fibrosis and cross-linking of advanced glycated end products (AGE) between myocardial fibrils and within the glomerular filtration apparatus. There may be an intermediate stage of normal systolic function with abnormal diastolic function that may be associated with an increase in myocardial and renal interstitial matrix collagen as well as the binding of AGE to myofibrils and nephrons. Cardiomyopathy in diabetic patients with diminished kidney function may be present for years before the detection of fibrosis by noninvasive testing [1], such that angiotensin/mineralocorticoid receptor blockade or angiotensin-converting enzyme inhibition have unanticipated benefits by interfering with extracellular fibrosis in heart and kidney [2].

Ventricular diastole refers to the time between aortic valve closure and mitral valve closure, and in descriptive terms has previously been described as divided into four phases [3]. Attempts to assess myocardial stiffness during these phases utilizing radioisotope scan, magnetic resonance imaging, or positron emission tomography have only been partially successful. Near-infrared spectroscopy may enable a better understanding of ventricular diastolic dysfunction in the development of heart failure with preserved ejection fraction, which represents 50% of hospital admissions for acute heart failure. Earlier diagnosis by imaging, biomarkers or endomyocardial tissue examination could lead to improvements in quality of life and reduced hospital admissions for the syndrome of recurrent heart failure [4].

## 2. Kidney Dysfunction Mechanisms Interact with Cardiac Dysfunction: Clinical and Experimental

Cardiomyopathy has been described in type 1 diabetes (DM1) [5,6,7,8], hypertension [9,10], and obesity [11,12] each of which may ultimately have a renal role in uremia-specific cardiomyopathy [13,14] (Table 1) through cardiorenal linkages involving the interdependence of myocardial relaxation/contraction and tubuloglomerular handling of fluid and uremic toxins. Therapies designed to limit the loss of cardiac and renal functional reserve have mainly focused on hemodynamic manipulations of preload, afterload, and regional perfusion variation. However, the heart and kidney are also linked by the non-hemodynamic effects of the intra- and extracellular renin-aldosterone angiotensin axis. The cellular effects of the renin-aldosterone angiotensin axis upon vascular and cardiomyocyte structure, function and distribution, in conjunction with the interstitium in the early stages of renal disease, have been underemphasized while many studies describe cardiorenal relationships late in failure of both organs.

Current 10-year risk calculators for cardiovascular disease may not provide measurements of renal function for risk estimation nor do they consider heart failure a defined clinical cardiovascular event [15]. Among 13,000 adult individuals seen repeatedly over a 6-year period for type 2 diabetes (DM2, mean age 65) 77% had been already diagnosed with hypertension, and 27% had an estimated glomerular function rate below 60 mL/min per 1.73 m^2^ of body surface area [16].

Both DM1 and DM2 may be complicated by capillary albumin leakage with sympathetic nervous system activation due to relative hypovolemia. Autonomic neuropathy with unopposed sympathetic nerve stimulation as part of the DM1 cardiorenal complex has been the focus of prior studies [17,18,19,20] that demonstrated reduction of left ventricular mass by intensive treatment of hyperglycemia and hypertension. Occasional reports describe the reversal of experimental high blood glucose with antihypertensive medications [21], suggesting crosstalk between metabolic and hemodynamic pathways. Table 1 illustrates mechanisms by which cardiac dysfunction may be associated with hemodynamic and toxic consequences of kidney dysfunction. These include effects of uremic toxins on myocardial dysfunction through fibrosis via indoxyl sulfate [22,23] and cardiomyocyte hypertrophy by fibroblast growth factor 23 [24], phosphate [25], beta 2 microglobulin [26] as well as amyloid deposition with inhibition of ventricular contraction/relaxation [27,28,29].

A mouse model, genetically altered to reproduce disruption of the glomerular basement membrane, has been used to study left ventricular diastolic dysfunction (Table 2). Two intermediates in the mechanism are the proinflammatory cytokine osteopontin and 2-oxoglutarate, an inhibitor of efficient energy production by the mitochondria [30]. In a separate mouse model with hypertension and insulin resistance provoked by aortic constriction, cardiomyopathy has also been demonstrated to be associated with limited mitochondrial production of ATP [31]. Similarly, the mouse model for DM1 (Akita) has demonstrated mitochondrial abnormalities in structure and function [32]. Altered cardiomyocyte contractility has been demonstrated in the db/db insulin-resistant mouse model to be associated with decreased calcium handling by the sarcoplasmic reticulum [33].

Evidence for ventricular dysfunction increases as renal function decreases even with normal coronary anatomy. Increased pulse pressure associated with left ventricular dysfunction [34] results from metabolic syndrome with resistant hypertension and unopposed sympathetic nerve activity. Post-mortem [35] and myocardial biopsy [36] studies confirm the presence of interstitial fibrosis with collagen fibers and hypertrophied disorganized cardiomyocytes on electron microscopy. With the development of severe renal dysfunction, increasing inter-cardiomyocyte fibrosis is noted. Fibrosis of the myocardial interstitium presents clinically as congestive heart failure with normal ejection fraction, atrial fibrillation, ventricular arrhythmia, or remodeling after infarction [37]. These processes progress with decreasing kidney function but even at the end-stage may be partially reversible after a successful kidney transplant. Although left ventricular mass estimates by echocardiography appear to be greater than those from magnetic resonance images, each has been useful [38] in uremia.

Uremic toxin removal by isovolemic hemodialysis improves ejection fraction and fiber shortening without blood pressure or diastolic volume change, demonstrating that these toxins impair myocardial function [39]. Two uremic toxins have been associated with cardiac hypertrophy: fibroblast growth factor 23 [40,41] and beta 2 microglobulin [26] (Table 3).

At each stage of renal dysfunction, the prevalence of cardiac arrhythmias and heart failure increases. In the pre-dialysis phase, fewer arrhythmias were noted among patients with advanced CKD on ambulatory EKG than among hemodialysis patients [42]. Among dialysis patients, demonstration of ventricular ectopy is associated with increased cardiovascular risk [43]. With diminished kidney function and associated increased left ventricular mass, prolongation of QT interval with increased QT dispersal has been documented [44]. Increased QT dispersal in the setting of unopposed sympathetic nerve activity potentiates arrhythmia risk. In the human heart transplant recipient, denervated organs respond to circulating catecholamines with elevated heart rate. This unopposed sympathetic activity presents a unique arrhythmia risk period in the initial 30-day period following heart transplantation [45]. Increasing duration of diabetes is likewise associated with loss of parasympathetic function and relatively unopposed sympathetic activity, potentiating arrhythmic risk.

Long-term echocardiographic observations of dialysis populations have noted a survival advantage when ejection fractions were greater than 45% [20] with higher rates of mid-wall systolic fractional shortening [46,47]. Levels of B natriuretic peptide associated with atrial dilation and heart failure rise exponentially with decreasing estimated glomerular filtration rate (eGFR). Levels of another marker for heart failure (HF), big endothelin, rise in a linear fashion as eGFR decreases. Investigators have been able to distinguish patients without HF from those with HF and preserved ejection fraction (HFpEF) with both markers, using reliable reference ranges for different levels of eGFR [48].

Patients with heart failure due to myocardial infarction exhibit diminished utilization of fatty acids until ventricular dysfunction has stabilized. Both myocardium [49] and injured proximal tubules in the kidney [50] can return to steady-state utilization of fatty acids after an efficient shift to glucose for short-term acquisition of energy. This may not always be possible in the insulin-deficient or resistant state.

A previously under-appreciated dysfunction of cardiac muscle calcium signaling involving the tricarboxylic acid cycle is being studied in both humans [51] and the db/db (leptin receptor-deficient) DM2 mouse [52]. Calcium uptake in mitochondria has been noted to be deficient in diabetic cardiomyopathy. Experimental increase in expression of a critical area of the inner mitochondrial membrane that regulates calcium uptake results in improved myocardial contraction. Myocardial relaxation is closely associated with the movement of calcium into the sarcoplasmic reticulum [53] (Table 2).

In kidney failure accumulating strands of collagen with or without cross-linking by AGE cause resistance to relaxation following contraction [54]. Over extended periods of time, the oxidation of fatty acids may be inefficient compared to that of glucose for energy utilization. Other saturated fatty acids (ceramides), which are not a source of energy production, have been associated with insulin resistance [55] and cardiovascular events [56,57].

Table 3 and Figure 1a summarize pathways to cardiac collagen accumulation through either transforming growth factor (TGF) beta [58,59,60,61] or tissue necrosis factor (TNF) alpha [62,63]. Endothelin is a mediator in this process. The role of TNF alpha in the rapid decline of renal function in diabetic patients with nephropathy [64] draws attention to interstitial fibrosis as a common mechanism of dysfunction in both the kidney and heart. Procoagulant factors plasminogen activator inhibitor (PAI) [65] and thrombospondin [66,67] are intermediates in the activation of TGF beta (Table 4, Figure 2). Activators of PAI include several factors that contribute to hypertension through vasoconstriction: angiotensin ll [68], aldosterone [69,70], phenylephrine [71], and norepinephrine [72]. Endothelin [58,59,60] and platelet-derived growth factor [58] may also contribute to this pathway.

Inhibition of the renin/angiotensin/aldosterone pathway by ACE inhibitors results in increased expression of bradykinin (BK), which promotes the degradation of collagen (Figure 1b). Collagen biomarkers may have predictive value for congestive HF with and without preserved ejection fraction [73]. Klotho (KL), the anti-aging gene, operates to prevent organ damage from uncontrolled hypertension [74], minimize vascular calcification in chronic kidney disease [75], and minimize fibrosis [76] induced by both TGF beta [77] and fibroblast growth factor 23 (FGF23) [78]. Table 5 suggests a role for the new calcimimetic agent etelcalcetide versus standard vitamin D analogue alphacalcidol to suppress parathyroid hormone (PTH) and FGF23, which have been associated with cardiac endpoints in patients with kidney disorders. Among hemodialysis patients (n = 62) treated for 12 months, a significant difference in reduction of LVH was found for etelcalcetide, which, in addition, was associated with reductions in PTH and FGF23 [79].

Although the sodium glucose transporter-2 inhibitor (SGLT2i) empagliflozin may be beneficial in the reduction of HFrEF, it was not found to alter the accumulation of collagen in an experimental non-diabetic mouse model [80]. Drugs that minimize pathological effects of thrombin, TGF beta, angiotensin II, and aldosterone on cardiac fibrosis may operate to a certain degree by inhibiting the expression of PAI. Figure 2 summarizes the pathogenesis of activation of PAI through mechanisms involving inflammation/hemostasis, insulin resistance, oxidative stress, and hypertension.

## 3. Obesity/Metabolic Syndrome/Diabetes/Hypertension and Cardiovascular Dysfunction

Obesity and hypertension are closely linked through autonomic centers in the midbrain. The satiety hormone leptin controls appetite in the normal population; leptin resistance is present among obese adults. The center for autonomic nervous system activation, which is hyper-stimulated during cycles of weight gain and under-stimulated during weight loss [81], is located in close proximity to the satiety center. Among leptin resistant (obese) patients, adiponectin levels are diminished. Among women, but not men, levels of adiponectin appear to be related to increased ventricular mass and diastolic dysfunction [82].

Even among normotensive populations, obesity is associated with excess aldosterone, which stimulates cardiomegaly [83]. Outcome research in 4189 mostly hypertensive elderly patients with HFpEF randomly treated with or without ACE inhibitors demonstrated a 9% reduction in hospitalization or mortality with the use of ACEI (Hazard Ratio = 0.91, *p* = 0.028) [84]. ACE inhibitors, angiotensin receptor blockers, beta-blockers, and statins have been shown to diminish interstitial fibrosis by inhibiting the activity of cardiac fibroblasts [85]. The response of obese hypertensive patients to diuretics is blunted among patients with heart failure when compared to hypertensive controls. Increases in cardiac filling pressure and renal venous flow impedance are associated with resistance to diuretics. Once atrial dilation begins, the expression of atrial natriuretic peptide correlates with diuretic effectiveness in reducing plasma volume [86].

A high-sucrose or high-fat diet impairs systolic and diastolic function [87] by reducing energy production through inhibition of cytosolic ATPase or mitochondrial electron transport complex II [88]. When the content of palmitate [89] or the sphingolipid ceramide is elevated in the cardiac sarcoplasmic reticulum, contractile dysfunction occurs through limited production of high-energy phosphate, associated with a shift from fatty acid to glucose oxidation [90]. Ischemia leads to the uncoupling of oxidation of fuels from the production of high-energy phosphate with the resultant release of calcium from mitochondria via permeability pores [91]. Cellular integrity is challenged by increased levels of calcium, lactic acid, hydrogen ion, and reactive oxygen [92].

The human heart normally generates up to 5 kg of ATP/day, amounting to an expenditure of its complete supply of ATP four times/minute. Fatty acid oxidation accounts for 70% of ATP production; glucose/lactate accounts for 30%. Since oxidation of glucose is more efficient than that of fatty acids, glucose oxidation may be recruited up to 40% of the total in times of cardiac stress. Among numerous steps in insulin post-receptor signaling, the Akt-1 position has a direct inhibitory effect on fatty acid oxidation. Down-regulation of fatty acid oxidation includes inhibition of the effect of carnitine palmitoyl transferase to convey palmitate into mitochondria and increased expression of uncoupling protein, both of which result in the diminution of ATP generation [93].

The chemistry of adipose tissue includes systems for the synthesis of several active participants in hypertension. Excess mineralocorticoid synthesis and excretion appear to accompany obese hypertensive patients in chronic kidney disease stages 2–4 and aging [94,95]. Inhibition of the mineralocorticoid receptor with spironolactone or eplerenone has been useful in controlling both fibrosis [95,96] and hypertension over 6 months with a reduction in mortality [97,98,99]. In addition to secretion from the adrenal glomerulosa zone, aldosterone may be excreted by adipose tissue through an intact renin-angiotensin-aldosterone pathway [100]. Adipose tissue may also give rise to a mineralocorticoid releasing factor [101] identified as an epoxy-keto derivative of linoleic acid [102].

Obesity is characterized by elevated levels of leptin and diminished adiponectin. Following weight loss after bariatric surgery [103,104] levels of reactive oxygen species of lipid origin are corrected [105,106]. Isotope-labeled imaging of utilization of long-chain fatty acids or glucose [107] is possible through the application of positron emission technology (PET scan). Magnetic resonance imaging is being applied to estimate rates of carbohydrate metabolism at the level of pyruvate/lactate interaction [108]. When cardiomyocytes under stress can’t beta-oxidize long-chain fatty acids quickly enough for urgent needs, disposal of glucose for energy occurs through the glycolytic and tricarboxylic acid cycles [109]. Restricted movement of fatty acid moieties such as palmitic acid through the inner wall of mitochondria by the enzyme carnitine palmitoyl transferase is seen as the relevant mechanism for inadequate energy production from fatty acid oxidation in ischemic hearts [46] and in the Zucker diabetic fatty (ZDF) rat heart [110]. To explore these fatty acid mechanisms, a trial of omega -3 fatty acids has been conducted in persons with and without type 2 diabetes [111].

Restriction of movement of misfolded proteins in association with ubiquitin from the cytosol into the endoplasmic reticulum of cardiomyocytes in the process of autophagy may lead to toxic accumulation of proteins in dilated cardiomyopathy with heart failure [112]. Toxic accumulation of fatty acids can alter cell signaling and promote apoptosis [113]. Very elevated levels of triglycerides associated with insulin resistance may be toxic through the inactivation of post-receptor insulin signaling at the point of Akt [114]. The ZDF rat develops elevated left ventricular levels of triacylglycerol before irreversible fibrosis. Although metformin and fenofibrate decrease ventricular triacylglycerol, only fenofibrate decreased fibrosis in this model [115]

## 4. Diabetes Mellitus with Congestive Heart Failure

A recent review illustrates correlations between diabetes and HFpEF, describing a heterogeneous group of phenotypes driven in part by comorbidities [116]. Irrespective of other comorbidities, readmission rates for patients with diabetes mellitus and HFpEF were higher, largely driven by heart failure readmissions, especially in the presence of diminished renal function [117]. This review supports decades of prior observations demonstrating interactions between diabetes and adverse cardiovascular consequences [116]. Such cohorts of patients, however, were not enrolled in studies until after their first hospital presentation with documented clinical heart failure, ignoring the elevated risk of initial events resulting from persistent hyperglycemia. An effort to assess the prevalence of heart failure related to DM2 demonstrated that when 271,174 individuals with DM2 were compared to 1,355,870 individuals without diabetes, heart failure prevalence was 45% higher (hazard ratio 1.45) in the diabetic cohort [118]. The trial of omega-3 fatty acids in persons with or without type 2 diabetes demonstrated significant protection from initial heart failure hospitalization [111] an effect that was stronger over a follow-up of 6 years in the Black population

While leakage of albumin into retinal macula has clinical significance, leakage through microvasculature in the heart and kidney may remain undetected until a more advanced state of disease [119] at which point SGLT-2 inhibitors may be effective in preserving function [120]. The biguanide metformin has been associated with the prevention of altered myocardial function among diabetic patients [121]. In the canine heart failure model, metformin is associated with phosphorylation of adenosine monophosphate kinase and endothelial nitric oxide synthase with improved function [122]. No studies of improved control of myocardial capillary leakage are available.

## 5. Diabetes Mellitus with Collagen Crosslinking Causing Myocardial Stiffness

Accumulation of advanced glycosylation end products (AGE) occurs with chronic insulin resistance and is associated with impaired ventricular relaxation (stiff ventricle). Simultaneously, renal glomerular filtration may be equally impaired through the same cross-linking of collagen molecules [53]. Attempts to diminish the accumulation of AGE [123] or to break collagen crosslinks [124] have not proved feasible clinically despite success in laboratory-based experiments [125]. Another pathological effect of AGE is increased expression of inflammatory signals, one of which may be nuclear factor kappa beta [126].

Heart failure admissions have been associated with a marker for collagen cross-linking: a low ratio of carboxy-terminal telopeptide to matrix metalloproteinase indicates resistance to collagen degradation by matrix metalloproteinase, resulting in ventricular stiffness [127]. Two AGEs (carboxy methyl lysine and pentosidine) have been associated with retinopathy in long-standing diabetic patients [128], but carboxy methyl lysine was not found associated with ventricular muscle collagen or myocardial contraction/relaxation pathology [129]. Experimental studies with AGE have demonstrated accumulation of methylglyoxal in chronic kidney disease [130] and in congestive heart failure in the myocardial infarction model [131]. Accumulation of AGE studies in wild-type and transgenic animals have been carried out in non-diabetic states [131] in which the enzyme glyoxylase was used to diminish concentrations of methyl glyoxal, resulting in less cardiac fibrosis. Congestive heart failure/dysfunction is associated with elevated levels of carboxy methyl lysine but is a less reliable predictor of diabetic nephropathy [132]. Elevated levels of autoantibodies to cardiac myosin in murine subgroups with A1c > 9% as opposed to <7% [133] are associated with enhanced expression of CD4+ T cells, which were profibrotic in the cardiac interstitium [133].

Increases in tissue collagen or circulating procollagen result in resistance to cardiac chamber filling or “diastolic stiffness”. Production of AGE resulting from hyperglycemia leads to cross-binding from one collagen branch to another with resistance in diastolic filling and arterial distension. In addition to binding across branches of peptide chains, AGE also attaches to lipids, which can be deposited in the myocardium with toxic effects. Among 125 DM2 study subjects with normal ejection fraction and no cardiovascular disease, matched for age and gender, non-invasive measurements revealed significant differences in cardiac diastolic function and peripheral vascular stiffness [134]. When subjects were divided into subgroups by A1c ≥ 6.5% (n = 88) versus < 6.5% (n = 40), the group with the higher A1c demonstrated slower velocities of mitral annular motion in diastole (*p* < 0.001) and systole (*p* < 0.05), suggesting chamber stiffness. The higher A1c group demonstrated an increased incidence of LV hypertrophy, associated with higher blood pressure and increased carotid-femoral pulse wave velocity.

In a nondiabetic left anterior descending artery occlusion porcine model, empagliflozin was compared to control over two months. Empagliflozin caused loss of glucose in the urine, forcing the failing heart to resume preferential oxidation of free fatty acids, ketone bodies, and branched-chain amino acids. Indices of heart failure included increased left ventricular mass, increased LV diastolic volume and increased LV end-systolic volume, decreased ejection volume and contractile reserve with dobutamine, as well as increased circulating levels of normetanephrine, B-natriuretic peptide, and troponin. Myocardial metabolism was monitored by cannulation of the coronary artery and coronary sinus. As glucose uptake decreased with empagliflozin, uptake of free fatty acids, ketone bodies (beta-hydroxy-butyrate), lactate, and branched-chain amino acids increased. Control animals demonstrated increased expression of lactate and pyruvate dehydrogenase, consistent with decreased glucose utilization. Empagliflozin treatment was associated with increased activity of succinyl CoA-oxoacid CoA transferase (key enzyme in ketone body oxidation); adenosine monophosphate kinase (regulator of cell metabolism), and carnitine palmitoyl transferase (essential for the movement of fatty acids into mitochondria for generation of ATP) [135].

Myocardial energy depletion in diabetes is related to limited microvascular uptake of the substrate and to dysfunction in its utilization [136]. Microvascular uptake of the substrate can be estimated through cardiac magnetic resonance for myocardial perfusion index (MPRI) as well as cardiac oxygenation through blood oxygen level-dependent signal intensity change (SI delta). Dysfunction in the utilization of substrate, associated with dysfunction of mitochondria, leading to impaired transfer of energy to myofibrils, can be estimated through phosphocreatine/ATP ratio. A total of 32 DM2 study subjects (mean A1c 7.4%) on oral agents matched with 17 controls underwent studies before/after either leg exercise or vasodilation with the infusion of adenosine. There were no instances of > 50% coronary obstruction on CT angiography or of late gadolinium enhancement for interstitial fibrosis on cardiac magnetic resonance. Phosphocreatine/ATP after leg exercise was not reduced in controls but fell by 12% in DM2 subjects whose levels were at rest were already 17% lower. Following adenosine, both myocardial perfusion and oxygenation were blunted relative to controls. Results of post-exercise phosphocreatine/ATP measurements correlated with measurements of mid-ventricular and longitudinal systolic contraction.

Another area where mechanisms of heart muscle dysfunction may differ for persons with or without a diagnosis of insulin resistance looks at calcium flow following ventricular contraction. Passive ventricular filling requires a resetting of myocyte electrical potential through calcium flow out of the reservoir (sarcoplasmic reticulum). Cardiomyopathy in diabetes is associated with decreased movement of calcium in diastole as demonstrated by studies in the obese Zucker rat model of type 2 diabetes and heart tissue samples of type 2 diabetes patients with preserved ejection fraction congestive heart failure [2]. Persons with similar preserved ejection fraction heart failure also have delayed delivery of calcium to cardiomyocytes due to a disorganized series of tubules used for calcium movement rather than the decreased release of calcium from storage [137].

## 6. Transforming Growth Factor Beta, Experimental/Clinical Relationship to Myocardial Fibrosis

Table 4 summarizes multiple factors that modify cardiomyocytes and myofibroblasts. Transforming growth factor beta (TGF beta) regulates endothelial/mesenchymal transition to a profibrotic phenotype through a receptor complex involving ligands from the TGF beta superfamily [138]. TGF beta signaling acts as a common pathway for mitogen-activated protein kinase, phosphoinositide 3-kinase pathway, and certain inhibitory micro RNAs. Stimuli of endothelial/mesenchymal transition, which converge with TGF beta signaling, include glucose, endothelin-1, angiotensin II, and AGE. Intermediates found in experiments connecting TGF beta with endothelial/mesenchymal transition include inhibition of fatty acid oxidation in mitochondria along with methylation of DNA specific for cardiac fibrosis. Oxidative stress associated with an excess of hydrogen peroxide as well as a deficiency of nitrous oxide is additive to TGF beta-induced endothelial/mesenchymal transition. The common pathway to cardiac and renal failure associated with the expansion of the extracellular matrix is considered irreversible once fibrosis has been detected. Studies have shown that the emerging myofibroblast continues to generate collagen as it differentiates through potentially reversible stages. Since the process requires activation by TGF beta (Table 2), this may be a point at which therapy may reverse the process of fibrosis. Hemodynamic unloading by left ventricular assist devices [139,140] or after coronary artery bypass surgery has been demonstrated to arrest remodeling in the myocardium [141]. Attempts have been made in experimental models to use monoclonal anti-TGF beta antibodies in db/db diabetic mice to arrest changes within the kidney [142]. An in vitro study has demonstrated a return of myofibroblasts in end-stage heart failure to a stage of diminished collagen production through inhibition of TGF beta [143]. While synthesis of collagen prevents myocardial rupture after infarction, in myocardium without regional infarction, excessive synthesis of collagen contributes to fibrosis and malfunction.

## 7. Collagen/Titin Contribute to Normal Heart Structure/Function or to Pathogenesis of Heart Failure: Experimental/Clinical Interaction with Compliance, Elasticity, Plasticity

Given the insidious development and progression of cardiomyopathy, indirect noninvasive quantitation of myocardial contraction and relaxation offers benefit, but requires a better understanding of cardiac physicochemical properties. Compliance measures how well tissue can conform to pressure and is measured as a change in volume/change in pressure. Elasticity, the reciprocal of compliance, describes how well tissue returns to its original shape when pressure is removed. Plasticity refers to the ability of living cardiac tissue to change its state in response to stimuli, depending upon the size, thickness, composition and perfusion of the heart muscle. Not all properties are measurable by available non-invasive testing. Early identification of pathologic changes in these physical properties may lead to early interventions to improve clinical outcomes. Ventricular compliance and elasticity indirectly can be measured noninvasively. Indirect measurement of plasticity, however, remains elusive as it depends upon constituents of wall composition (cardiomyocytes, extracellular matrix, ventricular shape, thickness, fibrous skeleton, valve pathology or perfusion). Attempts to define plasticity through deformation imaging radioisotope scan, magnetic resonance imaging, or positron emission tomography have been only partially successful.

Elasticity and contractility were studied in juvenile rats after banding of the pulmonary artery. Female rats survived longer, demonstrating lower levels of right ventricular fibrosis, and lower degrees of expression of the calcium/calcineurin cascade [144].

Diabetes-associated cardiomyopathy likely involves two large proteins, collagen and titin. Collagen, located within the interstitial matrix, can be cross-linked between chains of amino acids by AGE (from chronically elevated circulating glucose), resulting in resistance to both contractions in systole and relaxation in diastole. Titin, located in the cardiomyocyte sarcolemma, has an I region, capable of extension during ventricular diastolic filling, a property that limits resistance to the rhythm of relaxation/contraction. Energy for diastolic extension of the I band of titin derives from the insulin signaling pathway, involving phosphorylation via phosphatidyl-3-kinase. Decreased phosphorylation of titin through reduced activity of protein kinase G has been attributed to hyperglycemia [145]. Insulin deficiency, then, is an immediate cause of cardiac dysfunction, reversible with insulin, metformin or the epidermal growth factor, neuregulin-1 [146,147,148]. Insulin and neurregulin-1 have been shown to improve phosphorylation in the I band region of titin with a two-fold decrease in passive cardiomyocyte stress in the streptozotocin-diabetic APO E+ mouse- DM1 model [147] (Table 2).

## 8. Mineralocorticoid Receptor Antagonism: Experimental/Clinical

An important study connects cardiac function in women to diabetic kidney disease [149]. Cardiac MRI and PET scans ruled out prior infarct or current ischemia as well as late gadolinium enhancement, an indicator of myocardial interstitial fibrosis. Coronary flow reserve was significantly higher for women at rest, but not following adenosine. Women had a significantly greater increase in serum aldosterone than men following angiotensin infusion. In a prior study, this group had found a blockade of mineralocorticoid receptors with aldosterone to improve coronary flow reserve [150]. Diastolic function by echocardiogram for women correlated directly with resting myocardial blood flow and indirectly with coronary flow reserve. Endothelial deletion of mineralocorticoid receptors has been associated with the preservation of diastolic function in an experimental mouse model [151]. Clinical studies demonstrate increased hospitalization risk with HFpEF when coronary flow reserve and diastolic function are abnormal. Mineralocorticoid receptor blockade in diabetic subjects with HFpEF in the TOPCAT study was associated with a lower risk of cardiovascular events [152]. Blockade of the receptor for aldosterone limits turnover of extracellular matrix thereby promoting improved survival in congestive heart failure [153]. Activation of protein kinase C linked to diabetic complications has now been associated with cardiomyopathy [154].

Mineralocorticoid receptor antagonists may also improve cardiac contractility by ATP energy generation within muscle [155]. Female mice fed Western diets that increased fat mass and insulin resistance demonstrated an improvement in insulin sensitivity with either knock-out or inhibition of mineralocorticoid receptor with spironolactone [156], which would be expected to provide an improvement in the generation of energy from carbohydrates within the cardiac muscle. ACE inhibitors, by blocking the generation of fibrosis within the myocardium, may promote muscle contraction efficiency [157,158,159]. In a model of acquired type 2 diabetes, ZDF rats have responded to treatment with peroxisome proliferator-activated receptor-gamma agonists, demonstrating lower end-diastolic pressure through improved neuregulin-1 activity with improved myocardial glucose oxidation, resulting in more efficient contraction [160] (Table 2).

An increase in matrix glycoprotein associated with collagen precedes interstitial fibrosis [132]. During periods of oxidative stress, angiotensin II enhances the expression of TGF beta and Tumor Necrosis Factor Alpha (TNF alpha), contributing to myocardial dysfunction [161,162,163,164]. TGF beta-induced interstitial matrix accumulation [165] may be blunted by anti-angiotensin medications [166,167], carvedilol [168], and statins [169], thereby minimizing heart and kidney dysfunction. TNF alpha-induced cardiomyopathy may have an NF kappa beta expression mechanism independent of the inflammatory cascade [170] (Figure 1a).

Allopurinol has been reported to improve arterial blood flow through a nitric oxide-dependent endothelial system activated by acetylcholine [171]. This vascular relaxation effect may account for the regression of left ventricular hypertrophy in DM2 [172] and stage 3 chronic kidney disease [173]. In follow-up studies of chronic heart failure, allopurinol appears to contribute to the reduction in hospitalization and mortality [174,175]. The hypo-uricemic/anti-inflammatory functions of allopurinol had been demonstrated to slow kidney dysfunction in DM1 animal models and are still under investigation in humans as the PERL study [176], which has had an initial report of non-significant impact on renal function [177].

The non-steroidal mineralocorticoid receptor agonist finerinone and SGLT2 inhibitors have shown positive effects on renal and cardiac outcomes in patients with DM2. Investigators now looking at the combination of the two drugs. Investigators studied the effects of a low dose combination of fenerinone and empagliflozin on cardiorenal outcomes in hypertensive and proteinuric transgenic rats (mRen)27Rats2. Endothelial dysfunction was induced by including nitrogen monoxide synthase inhibitor N (ω)-nitro-L-arginine methyl ester in drinking water [178]. All rats were pretreated with a diet including captopril 300 mg/kg in food. Rats treated with the low dose combination showed a reduction in urine protein, significantly reduced levels of serum creatinine and uric acid as well as significantly reduced systolic blood pressure compared to the monotherapy group and experienced a survival benefit compared to the placebo group. On histologic examination, hearts and kidneys from rats treated with a low dose combination showed less myocardial degermation and glomerulopathy and tubular atrophy. The low dose combination also demonstrated less cardiac and renal fibrosis than monotherapy.

## 9. Congestive Heart Failure with Preserved Ejection Fraction 

A retrospective analysis of 232,656 patients (Get With the Guidelines-Heart Failure) demonstrated that persons with diabetes treated for heart failure remained in hospital longer, were directed to a rehabilitation center more often or were more likely to be readmitted for heart failure within 30 days [179,180]. A review of prospective studies of HFpEF demonstrated that cohorts with diabetes have a greater risk than cohorts without diabetes for heart failure hospitalization or cardiac death [181]. Outcome studies comparing populations with diabetes to populations without diabetes have included use of candesartan (CHARM) [182]; digitalis (DIG) [183]; phosphodiesterase -5 inhibitor, sildenafil (RELAX) [184]; irbesartan (I-PRESERVE) [185]; and spironolactone (TOPCAT) [186].

Studies evaluating cardiovascular safety and efficacy of sodium-glucose transport inhibitors have revealed a lower incidence of cardiovascular events with empagliflozin (EMPA-REG) [187] and canagliflozin (CANVAS) [188], providing a convenient method for reducing both blood glucose and blood volume, thereby alleviating cardiac stress in patients with DM2 with adequate kidney function. The target population for use of SGLT 2 inhibitors will continue to increase as research investigates ketone bodies, which require less oxygen per molecule of ATP generated than long-chain fatty acids like palmitate or intermediate chain carbohydrates like glucose. In addition, arterial vascular injury repair may be enhanced with the use of SGTL2 inhibitors like dapagliflozin when ketone body concentration increases [188]. A study of persons with type 2 diabetes found dapagliflozin, which increases non-oxidative glucose disposal, and also decreased glucose oxidation while increasing fatty acid oxidation [189]. This interesting mechanism might contribute to returning the failing myocardium back to its natural fuel while eliminating excess glucose by kidney disposal.

Renal proximal tubule losses of glucose, sodium, and fluid with empagliflozin [190] and dapagliflozin [191] did not result in the generation of vasoactive neurohumeral hormones which would contribute to loss of plasma volume. Thus, the use of loop diuretics (furosemide, bumetanide) would be expected to render an additional benefit for individuals with congestive heart failure with or without preserved ejection fraction [190,191]. For the patient with type 2 diabetes preserved ejection fraction heart failure, sotagliflozin, an inhibitor of sodium glucose transport 1+ 2, has been demonstrated to improve cardiovascular outcomes [192]. The mechanism appears to be activation of sodium calcium transport as sodium glucose transport is inhibited by SGLT1. Improved intracellular calcium is the key feature [193]. Energy for skeletal muscle contraction may be mediated by enhanced insulin signaling thru the second messenger pathway (AMPK) to the glucose transport system (GLUT) [194]. There may be a beneficial effect of SGLT 2 inhibitors on cardiac metabolism [195] to prevent congestive heart failure. One hypothesis might be that when fatty acid oxidation is too slow and glucose oxidation is too fast, SGLT2 inhibitors are useful through the modulation of the delivery of ketones [196] and glucose.

The PARADIGM HF study of DM2 [197], using the combination of valsartan with the neutral endopeptidase neprilysin, (sacubitril), was associated with both a decrease in procollagen peptide (synthesis) and an increase in matrix metalloproteinase (degradation) in HFrEF. Current studies on the HFpEF population are underway.

Researchers have examined the effects on preservation of heart muscle anatomy and function by use of GLP-1 receptor agonists versus dpp4 inhibitors in a mouse model of cardiomyopathy in type 2 diabetes. Older mice fed a high-fat diet and subjected to constriction of the aorta develop an increase in ventricular mass/volume by echocardiography as well as increased end-diastolic pressure, eventually cardiac interstitial fibrosis. A very specific inhibitor of dipeptidyl peptidase was associated with all of these outcomes to a modest degree. By contrast, liraglutide produced no such changes [198]. A prospective study of 139 study subjects with DM2 using liraglutide (n = 45), sitagliptin (n = 49), and linagliptin (n = 45) achieved improved systolic/diastolic blood pressures in addition to a significant lowering of fasting/post-prandial glucose over 48 months of follow-up. Echocardiography was used to measure left atrial size. Doppler flow studies documented left ventricular diastolic filling pressure by measurement of left atrial emptying through analysis of septal mitral annular flow velocity (E/e′). Liraglutide demonstrated improvement while sitagliptin and linagliptin did not [199].

## 10. Hypertension Linked to Kidney Disease Rather Than Obesity or Diabetes Mellitus

Investigation into the relationships between left ventricular structure/function and hypertension with or without fluid overload is ongoing. For patients with both hypertension and kidney disease, increased ventricular mass due to obstructive arterial disease is associated with decreased survival [200]. Reduction in left ventricular mass is possible with intensive therapy [201]. Further studies on a patient with chronic kidney disease treated with dialysis or transplantation will be needed to identify the unique benefit of intensive therapy demonstrated through imaging [202] and laboratory biomarkers [203]. End-stage kidney disease patients on maintenance dialysis have longer survival if their body mass index is in the range of 25–35 kg/m^2^ [204], that is, a higher compared to a lower quintile. Among chronic dialysis patients, concentric left ventricular hypertrophy was associated with a lower prevalence of cardiovascular events than eccentric hypertrophy and was more responsive to size reduction with angiotensin-converting enzyme inhibition [204,205]. The explanation for this finding may be related to the relative impact of ACE inhibition on muscle as opposed to collagen, which is found in higher serum [205] and arterial tissue [206] concentrations in hypertension. Collagen synthesis as measured by levels of mRNA and protein may be delineated in endo-myocardial biopsies of hypertensive heart disease with or without heart failure [207]. Inhibition of ventricular contraction/relaxation due to collagen/fibrosis or amyloid deposition [27,28,29], can be quantitated by Doppler indices [208]. A unique situation occurs in persons who have had chronic kidney disease treated with ACE inhibition, and maintenance dialysis but who retain a functioning arteriovenous fistula (AVF) on immunosuppression after kidney transplantation. Those randomized to AVF ligation versus those randomized to no AVF ligation demonstrated a significantly greater reduction in LV mass by cardiac magnetic resonance imaging at both 6 months and five years [209]. This is most probably a manifestation of reduced blood pressure [210] with associated hemodynamic gains from a combination of improved kidney function plus ligation decreasing demands on cardiac output versus improved kidney function alone.

Diastolic dysfunction by clinical examination, chest x-ray, and transthoracic echocardiography was diagnosed in 11 of 190 (5.8%) candidates for live-donor kidney transplantation [211]. Removal of uremic toxins as well as excess salt/fluid in addition to correction of anemia by means of deceased-donor kidney transplantation was associated with a significant risk reduction (RR) of congestive heart failure at three years. Of the 67, 591 recipients a RR of 54% was recorded for candidates of normal body mass index (BMI) while the RR for candidates of significantly elevated BMI was 32% [212]. However, a BMI of 30 kg/m^2^ is associated with better outcomes than a BMI of 20 kg/m^2^ in the cohort [204].

There is an interest in biomarkers for heart failure in the cardiomyopathy population which usually involved obesity, hypertension, and diabetes. Biomarkers for cell proliferation are associated with HFrEF fraction while biomarkers for HFpEF are associated with inflammation [213]. Patients with diabetes expressed clusters of biomarkers for inflammation and fibrosis not seen in the non-diabetic patient population [212]. Further pathophysiologic differentiation of acute myocardial response to hypoxic stress will require measurements currently only available in research laboratories [214].

## 11. Role of Dysautonomia in Myocardial and Renal Adaptation to Stress

Cardiac autonomic dysfunction is associated with a more rapid progression of kidney dysfunction [17,215]. Obese patients with or without heart failure have evidence of neuro-humoral dysfunction. Treatment modalities that decrease this dysfunction have demonstrable short and long-term benefits [19]. Excessive sympathetic neuro-humoral activation has adverse consequences for both cardiovascular and renal function [81,216,217]. Since many patients with metabolic syndrome may have already reached the point of depressed parasympathetic function [20], unopposed sympathetic activity associated with hypertension is particularly prevalent in diabetes cohorts with early renal disease [18,19,217]. Unopposed sympathetic activity associated with hypertension may play a role in a higher prevalence of left ventricular hypertension and heart/renal failure [81]. Similarly, evidence for parasympathetic dysfunction is associated with a higher prevalence of progression of renal dysfunction, even in diabetes cohorts with early renal disease [17,18,217]. Cardiovascular risk has now been reported with multiple clinical states associated with an unstable autonomic nervous system [218,219,220]. Individuals with obesity-related sleep apnea may demonstrate hypertension from unopposed sympathetic activity [81], which would be of concern as an unsuspected cause of cardiovascular events [220]. In an exploratory analysis of the EMPA-REG Outcome trial, a cardiovascular/renal benefit has been reported in study subjects with sleep apnea [221]. The mechanism proposed is an increased renal excretion of glucose, resulting in a shift in energy production to fatty acid oxidation with an expected lower production of CO_2_, associated with diminished reflex constriction of the pulmonary artery [222].

Attempts to reduce systolic blood pressure with interruption of sympathetic nerve signaling either above or below the diaphragm have been reported. Renal sympathetic denervation via renal artery catheter has at times been an effective treatment for selected individuals [216]. Cardiac autonomic neuropathy has been connected with cardiomyopathy through the measurement of myocardial flow reserve by means of both PET and CT scans [223]. Cardiac autonomic neuropathy was associated with diminution of myocardial blood flow reserve and with ventricular dilation in DM1 nephropathy patients.

Cardiac autonomic function correlates with glomerular filtration rate. The normal balance of parasympathetic/sympathetic innervation accounts for increased heart rate (shorter R-R interval) with increased intrathoracic pressure during inspiration versus decreased heart rate (longer R-R interval) with decreased intrathoracic pressure during expiration. Increased intrathoracic pressure is associated with decreased venous return to the right ventricle with a reflex increase in rate to sustain blood flow to the brain. Persons with diabetes mellitus (DM) may have loss of parasympathetic innervation resulting in a loss of variation in the length of the R-R interval during the respiratory cycle [224,225]. During the examination of individuals with DM1, the incidence of unopposed sympathetic innervation may be significantly higher than anticipated on clinical examination [226]. On repeated examination with ambulatory ECG, individuals with albuminuria and progressive loss of glomerular filtration rate have a growing incidence of loss of parasympathetic cardiac innervation with a significant relationship to control of blood pressure [227] and blood glucose [228].

## 12. A Cardio-Renal Syndrome or a Simple Concordance of Multiple Disease States?

Just as kidney failure may affect cardiac hemodynamics, failure of the heart to provide adequate flow to the kidney will impair kidney function. Efforts have been made to develop a pathophysiologic taxonomy to classify various types of cardio-renal failure to provide a scaffolding for future research [119,229]. Metabolic syndrome and cardio-renal syndrome are closely interwoven. Large-scale population studies [17] cannot be expected to analyze resistance to insulin [18,230] or resistance to blood pressure medications by non-invasive estimations of stiffness of the aorta [231]. Risk factors for both cardiac and renal dysfunction are similar (hypertension, diabetes, smoking). The connection between left ventricular hypertrophy, distal neuropathy and kidney dysfunction associated with genetically mediated deposition of transthyretin amyloid synthesized in the liver has led to therapeutic advances [223,229]. For the majority of cardiorenal syndrome variants, current therapy will continue to rely upon concern for vascular targets of hemodynamics. Table 5 summarizes therapies as Standard of Care and Not Standard of Care. A candidate mechanism for future research might be members of the reduced nicotine-adenine phosphate (NADPH) family in its oxidized form (NOXN). These mechanisms have been identified in terms of aging-related hypertension [232] depending upon NOX activity associated with angiotensin 2 [233,234] and with aldosterone [235]. A renal relationship with fibronectin deposition in the glomerular mesangial matrix has been associated with NOX activity [236,237,238,239].

The ongoing COVID pandemic is associated with a new form of cardiomyopathy not entirely reversible for individuals recovering from infection, usually demonstrating additional complications in the pulmonary or renal systems. Elevated troponin levels at the time of COVID infection may be associated with higher mortality [240]. High titers of virus in interstitial spaces and macrophages, but not cardiomyocytes, have been found at autopsy [241]. The mortality rate for COVID-positive individuals with HFpEF was five-fold greater than for COVID-negative patients with HFrEF [242]. Since reports have identified diabetes mellitus, hypertension, kidney injury and kidney disease [243,244,245] as underlying conditions for mortality risk with COVID-19 infections, studies of relationships for long-term survivors will be needed. The acknowledgment that COVID-19 may be associated with acute and chronic kidney injury that may also not be reversible lends credence to the possibility of common triggers of myocardial and renal damage [246,247,248].

More work needs to be done to understand this connection. The DARE-19 trial, an investigator-initiated trial, looked at whether dapagliflozin can reduce the incidence of cardiovascular, renal, respiratory complications, all-cause mortality, or improve recovery in 1250 patients hospitalized with COVID-19 but not critically ill on admission. Patients were included if they had one or more cardiometabolic risk factors for complications of COVID-19 and were randomized to dapagliflozin 10 mg or placebo [247]. The risk factors include DM2, atherosclerotic cardiovascular disease, heart failure and chronic kidney disease. Mechanism of action is thought to include a shift to increased fatty acid oxidation and reduced reliance on glucose, anti-inflammatory properties with reduced C-reactive protein and interleukin-6 levels as well as decreased activation of NLRP3 inflammasone. Despite the promising possibilities based on the mechanism of action, the DARE-19 study did not show a statistically significant effect on primary or secondary endpoints, in patients with eGFR less than, equal to, or greater than 60 mL/min per 1.73 m^2^ or with acute kidney injury [248].

## 13. Conclusions

The pathologic interplay between heart and kidney dysfunction now classified as the cardio-renal syndrome demands a more nuanced understanding of the underlying disruption of common cellular pathways in the myocyte and the kidney (Figure 3). A better understanding is needed of the effects of diabetes, obesity, inflammation, and dysregulation of the sympathetic nervous system on pathologic changes in cardiac function prior to irreversible alterations in structure/function. While volume overload and hypertension will still need to be treated to limit the effects of the syndrome, clinicians need newer therapies to interrupt fibrosis at the cellular level as well as mechanisms that disrupt energy generation at the mitochondrial level.

## Figures and Tables

**Figure 1 ijms-23-07351-f001:**
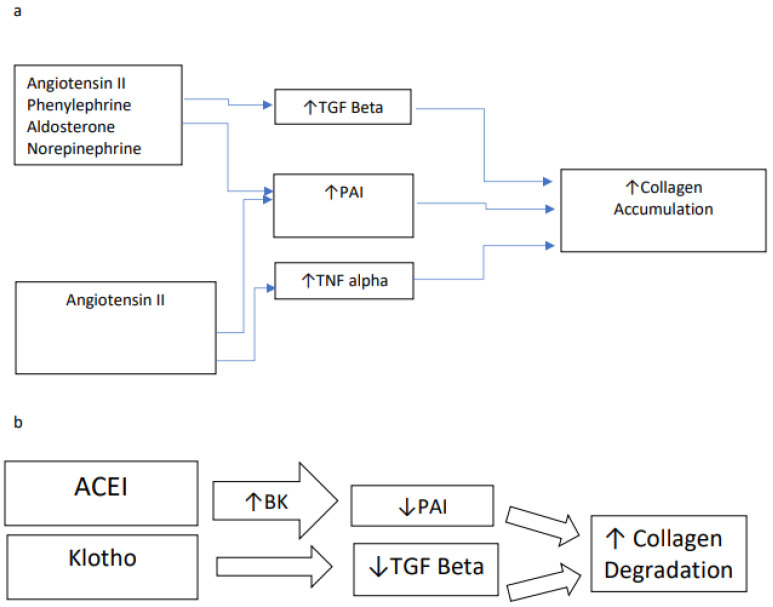
Metabolism of Collagen. ACE-I angiotensin converting enzyme inhibitor; BK bradykinin; PAI plasminogen activator inhibitor; TGF beta transforming growth factor beta; TNF alpha tissue necrosis factor alpha.

**Figure 2 ijms-23-07351-f002:**
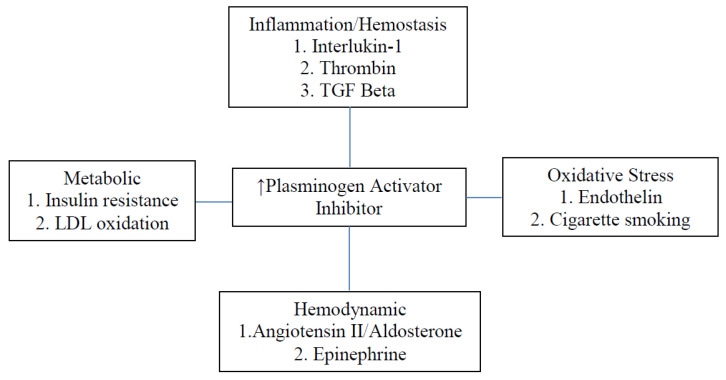
Increased Expression of Plasminogen Activator Inhibitor.

**Figure 3 ijms-23-07351-f003:**
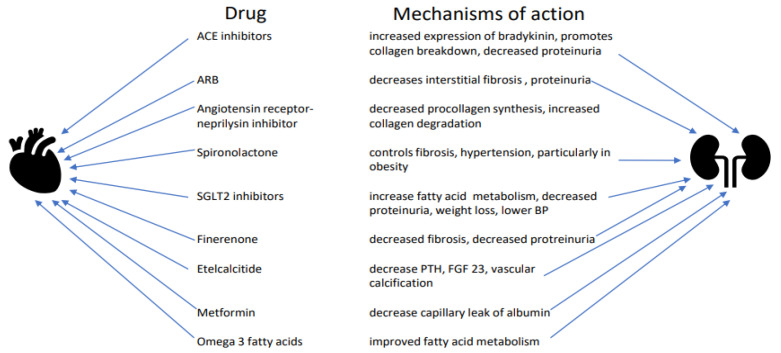
Drug actions in cardiorenal syndrome.

**Table 1 ijms-23-07351-t001:** Mechanisms of cardiac dysfunction that may interact with mechanisms of kidney dysfunction.

A. Increased systolic pressure with increased pulse pressure
B. Resistant hypertension in obesity due to central stimulation of sympathetic nervous system with local innervation via renal arteries
C. Increased collagen cross-linking with advanced glycated end-products
D. Uremic toxin inhibition of ventricular contraction through fibrosis (angiotensin II, aldosterone and indoxyl sulfate)
E. Uremic toxin stimulation of cardiomyocyte hypertrophy (phosphate, aldosterone, fibroblast growth factor 13, beta 2 microglobulin and indoxyl sulfate)
F. Amyloid deposition with inhibition of ventricular contraction/relaxation and renal glomeruler filtration

**Table 2 ijms-23-07351-t002:** Animal Models for Cardiomyopathy.

Disease	Model	Animal
Hypertension	Spontaneous hypertension	rat
	5/6 nephrectomy	rat
Heart failure	Right ventricular pacing	dog
	Aortic constriction	mouse
	Post-infarct	rat
Obesity	Leptin deficient, prone to DM2	ob/ob mouse
	Leptin-receptor deficient prone to DM2	db/db mouse
	Zucker obese prone to DM2	rat
Diabetes 1	Streptozotocin	rat
	Akita	mouse
Diabetes 2	Low dose streptozotocin	Wistar rat
	Transgenic	mouse
	OVE 26 with or without antioxidant protein, metallothionine	
	Long chain acyl synthase	
	Over expression of peroxisome proliferator activator receptor	

**Table 3 ijms-23-07351-t003:** Multiple Intermediate Factors in Cardiomyopathy Through Hypertrophy and Fibrosis.

Myofibroblasts	TNF-α, TGF-β, angiotensin II, aldosterone, endothelin, phosphate, parathyroid hormone, indoxyl sulfate	
Cardiomyocytes	Insulin pathways	Insulin receptor
		Insulin signal phosphatidyl inositol-3 kinase
		Insulin like growth factor receptor
	Pathways indirectly related to inflammation	Angiotensin II
		Endothelin
		Mammalian target of rapamycin
	Uremic toxin	Indoxyl sulfate

**Table 4 ijms-23-07351-t004:** Increased Expression of Plasminogen Activator Inhibitor.

Inflammation/Hemostasis	Interleukin 1
	Thrombin
	TGF Beta
Metabolic	Insulin resistance
	LDL oxidation
Oxidative stress	Endotoxin
	Cigarette smoking
Hemodynamic	Angiotensin II/aldosterone
	Epinephrine

**Table 5 ijms-23-07351-t005:** Therapy in Cardiomyopathy of Diabetes with Hypertension + RenalDisease.

**Standard of Care**	
Angiotensin	Angiotensin converting enzyme inhibitors
	Angiotensin receptor blockers
	ARB (valsartan) +Neprilysin inhibitor (sacubitrirl)
4Mineralocorticoid receptor agonist	Spironolactone, eplerenone
Sodium glucose transporter 2 inhibitors	Empagliflozin, dapagliflozin, canagliflozin
**Not Standard of Care**	
Limit myocardial capillary leakage of albumin	Metformin
Antifibrosis	fenofibrate
Improved control of body fluid level	SGLT1 and 2 inhibitors
Left ventricular hypertrophy reduction	Calcimimetic etelcalcitide

## Data Availability

Not applicable.

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
