# Peer review of "The Diabetic Cardiorenal Nexus"

_ijms, 2022, doi:10.3390/ijms23137351_

Round 1

Reviewer 1 Report

D’Elia et al. summarize the role of metabolic risk factors, including hyperglycemia, diabetes mellitus, and obesity in the development of chronic kidney disease and heart failure (i.e., cardiorenal syndrome) in their review article. I was the Reviewer on the first and second submissions of this MS to the IJMS. The authors have substantially revised their MS and significantly improved its quality compared to the previous versions. The only thing I miss is a figure summarizing the most important mechanisms and therapeutic targets in cardiorenal syndrome with DM or metabolic syndrome. A major spelling check is also needed. In summary, I suggest the approval of the MS for publication.

Author Response

We thank the reviewer for the comments. We will go through the article carefully to correct spelling mistakes, typing errors.

In terms of figures, Figures 1 and 2 along with table 5 summarize the pathways discussed in the article and describe the limited therapeutic agents available to intervene. We had added a figure three summarizing the action of various drugs in cardiorenal syndrome.

Reviewer 2 Report

This is an excellent and very detailed review of the various mechanisms that connect renal and cardiac dysfunction in the context of diabetes, obesity, and hypertension. The authors attempt to associate preclinical data with the clinical implications of available treatments (such as SGLT2i) and they indeed manage to provide a complete overview of the topics discussed. 

I only have a few comments:

1. The review seems to lack a consistent internal structure and the authors often mix the results of animal and human studies. I would suggest that the content can be organized on the basis of the nature of available evidence, i.e. first the results of animal studies and clinical / human data to follow in each section.

2. There are some fascinating new data on a novel non-steroidal MRA, finerenone, which from a pathophysiological perspective could interfere with some of the mechanisms involved in the dysregulation of the cardiorenal axis discussed in the review. I think it is worth discussing these data.

3. Page 15, line 3: there is a small typo in the word concordance (one additional “c”)

4. Page 15, section “A cardio-renal syndrome or a simple concordance of multiple disease states”: In this section, the authors very cleverly refer to COVID-19. Indeed, it has been suggested that the cardiorenal damage caused by COVID-19 shares common mechanisms with those that lead to diabetes complications. In this context, it is worth mentioning the rationale behind using SGLT2i in COVID-19 and the mechanisms of actions of gliflozins that in theory can justify a benefit (https://doi.org/10.1007/s11096-021-01256-9).

Author Response

We thank the reviewer for the comments. We have tried to organize the article along the lines suggested by the reviewer in terms of animal and human studies. emphasizing whether the outcome was histological or clinical. Often the discussions starts with a clinical outcome and then looks at the mouse models as a possible explanation for the clinical model.

We have included a brief discussion of the role of finerenone in the cardiorenal syndrome and expanded the discussion about COVID 19, similarities to diabetic kidney disease and possible role for SGLT2 inhibitors with negative outcome in one human trial with dapapliflozin.

This manuscript is a resubmission of an earlier submission. The following is a list of the peer review reports and author responses from that submission.

Round 1

Reviewer 1 Report

D’Elia et al. summarize the role of diabetes mellitus in the development of cardiorenal syndrome in their review article. I was the Reviewer on the first submission of this MS to the IJMS. In this second submission, the authors did not show any intention to improve the bad quality of the Tables and Figures and find the focus in the text. Therefore, I can not support the further review process of this article in the present form. 

I did not get answers to my original questions or see improvement in the criticized parts of the MS.

My previous requests and suggestions were the following:

"The topic is novel and interesting; however, the MS seems to be fuzzy and unfocused. Please find some suggestions and requirements to improve the quality of the MS and help the understanding of the readers.

Major comments:

  1. It is tough to follow the chapters of the MS because preclinical and clinical observations are mixed. Therefore, it isn't easy to find the focus of this MS. Several chapters and Tables do not help the understanding (e.g., Compliance, elasticity, plasticity, or Table 4). This reviewer suggests reorganizing the MS as follows: 1) Definition and epidemiology of cardiorenal syndromes. Please clarify which types of cardiorenal syndrome will be discussed in this review (I think the main focus is on type 4 cardiorenal syndrome in DM), 2) Clinical significance, and the manifestation of cardiorenal syndrome. 3) Major (patho)physiologic factors in the crosstalk between heart and kidney failure in DM:  kidney disease-specific and non-specific factors, 4) Molecular mechanisms and interactions in the development of cardiorenal syndrome: 4/a) Hypertension, 4/b) Obesity, 4/c DM forms, 4/d Heart failure forms. It would be necessary to show a summary figure on the molecular mechanisms and interactions. 5) Drugs used to improve cardiac and/or renal failure in DM. Please summarize the data of these experiments in a Table (Type of animal model, the drugs with dose and endpoints). Are there articles investigating both the cardiac and renal function in DM animal models? 6) Clinical observations and trials to improve the condition of DM patients suffering from cardiorenal syndrome. Summarize the data in a Table. 7) Conclusions with a Figure showing the therapeutic targets of cardiorenal syndrome in DM.
  2. Tables 1-2. It would be more helpful for the readers to visualize and summarize the content and mechanisms of Tables 1 and 2 on Figures. 

Minor comments: 

  1. Abstract: The second part of this sentence seems to be a fragment: "Heart failure hospital admissions and readmission often occur, however, in the presence of metabolic, renal dysfunction and relatively preserved systolic function."
  2. References are missing in the text (lines 45-55). (If this chapter remains in the text).
  3. Line 112: the authors might want to write a genetically altered experimental mouse model instead of "an experimental mouse genetically altered."
  4. Line170: ambulatory EKG. Please correct it ambulatory ECG.
  5. Table 3 is after Table 4.
  6. Figure 1a: please correct "norepineprine" to norepinephrine."

Reviewer 2 Report

There are a number of important synergistic interactions between heart disease and kidney disease. The interaction is bidirectional, as acute or chronic dysfunction of the heart or kidneys can lead to acute or chronic dysfunction in the other organ. Current review is discussed several studies based on the animal models and patients with diabetes, other comorbidities. Authors trying to illustrate the clinical and pathophysiological importance and their associations between the cardiac and renal dysfunction, suggesting that this review will give insights into identify the several targets which allow the treatment and prevention of cardiorenal syndrome. However, this manuscript did not fully/deeply describe the potential mechanisms involved and casual relationship between the heart and Kidney dysfunction. There are several major concerns needs to be corrected. The flow is missing, could not really convince the involvement of potential mechanisms and targets in this manuscript. Also the tables and Figures were not properly addressed. Some of the mechanisms discussed in this manuscript has lack of experimental evidences. Although the authors mentioned about the animal models of cardio renal syndrome, could not discussed about major findings from these models. There is also important animal model missing like DOCA-induced hypertensive animal model that develop both cardiac and renal fibrosis. Authors also discussed about the Covid related heart and kidney dysfunction which do not have any major evidences. Based on the extensive review analysis, this manuscript is not suitable for publication and out of scope in IJMS.